# Predefined and data-driven CT radiomics predict recurrence-free and overall survival in patients with pulmonary metastases treated with stereotactic body radiotherapy

**Pascal Salazar**[1], **Patrick Cheung**[2], **Balaji Ganeshan**[3], **Anastasia Oikonomou**[4]*

**1** Canon Medical Informatics, Minnetonka, MN, United States of America, **2** Department of Radiation Oncology, Sunnybrook Health Sciences Centre, University of Toronto, Toronto, Ontario, Canada, **3** Institute of Nuclear Medicine, University College London, London, United Kingdom, **4** Department of Medical Imaging, Sunnybrook Health Sciences Centre, University of Toronto, Toronto, Ontario, Canada

* anastasia.oikonomou@sunnybrook.ca

**Data Availability Statement:** https://github.com/psalaz83/Metastases_Radiomics https://github.com/psalaz83/Metastases_Radiomics/blob/main/README.md.

## Abstract

### Background

This retrospective study explores two radiomics methods combined with other clinical variables for predicting recurrence free survival (RFS) and overall survival (OS) in patients with pulmonary metastases treated with stereotactic body radiotherapy (SBRT).

### Methods

111 patients with 163 metastases treated with SBRT were included with a median follow-up time of 927 days. First-order radiomic features were extracted using two methods: 2D CT texture analysis (CTTA) using TexRAD software, and a data-driven technique: functional principal components analysis (FPCA) using segmented tumoral and peri-tumoural 3D regions.

### Results

Using both Kaplan-Meier analysis with its log-rank tests and multivariate Cox regression analysis, the best radiomic features of both methods were selected: CTTA-based "entropy" and the FPCA-based first mode of variation of tumoural CT density histogram: "F1." Predictive models combining radiomic variables and age showed a C-index of 0.62 95% with a CI of (0.57–0.67). "Clinical indication for SBRT" and "lung primary cancer origin" were strongly associated with RFS and improved the RFS C-index: 0.67 (0.62–0.72) when combined with the best radiomic features. The best multivariate Cox model for predicting OS combined CTTA-based features—skewness and kurtosis—with size and "lung primary cancer origin" with a C-index of 0.67 (0.61–0.74).

**Funding:** The author(s) received no specific funding for this work.

**Competing interests:** I have read the journal's policy and the authors of this manuscript have the following competing interests: PS is an employee of Canon Medical Systems. BG (who was not a data controller/processor for this study) is the co-founder/co-inventor of TexRAD texture analysis software used in this study and a shareholder (not an employee) of Feedback Plc., a UK-based company which owns, develops, and markets the TexRAD texture analysis software. PC declares no competing interests. AO declares no competing interests.

**Abbreviations:** RFS, recurrence free survival; OS, overall survival; SBRT, stereotactic body radiotherapy; CTTA, CT texture analysis; FPCA, functional principal components analysis; TDT, tumor doubling time; LR, local recurrence; PTV, planning target volume; AR, any recurrence; FVE, fraction of variance explained; ROI, region of interest; SSF, spatial scale filter; SD, standard deviation; MPP, mean of positive pixels; RMST, restricted mean survival time; K-M, Kaplan-Meier; SABR, stereotactic ablative radiotherapy.

## Conclusion

In conclusion, concise predictive models including CT density-radiomics of metastases, age, clinical indication, and lung primary cancer origin can help identify those patients with probable earlier recurrence or death prior to SBRT treatment so that more aggressive treatment can be applied.

## Introduction

The lung is one of the most frequent first sites for cancer recurrence, harboring 36%–42% of initial metastases [1]. Identifying in-vivo biomarkers of lung metastases with more favorable outcomes would benefit the selection of patients for more aggressive treatments. However, the reported clinical and imaging biomarkers associated with the worst prognoses in metastatic diseases are scarce in the literature. Tumour doubling time (TDT) is significantly associated with the prognosis in 5-year post-metastasectomy survival [1], but this biomarker requires a 2–3 month follow-up and serial scans, precluding early patient stratification. Moreover, tumour size and morphologic criteria from RECIST guidelines do not always reflect the effect of the treatment in metastatic cancer [2]. Other clinical variables affecting patient survival or the recurrence of pulmonary metastases, such as the age, sex, lesion location, or the origin of the primary cancer are poorly documented [3,4]. In this context, radiomic-based CT texture analysis to assess tumour heterogeneity has attracted much of the researchers' attention [5]. Intra-tumoural phenotype heterogeneity is known to be associated with poor outcomes in terms of overall survival or recurrence in various types of primary tumours such as esophageal cancer [6], non-small cell lung carcinoma [7], head and neck squamous cell carcinoma [8] and colorectal tumours [9].

Moreover, pretreatment survival analysis using clinical and radiomics variables has shown encouraging results for prognostication of lung cancer treated with definitive chemoradiation therapy [10] or stereotactic body radiotherapy (SBRT) [11] and for pancreatic cancer treated with surgical resection [12], distant metastases in lung adenocarcinoma [13], and liver metastases in colorectal cancer treated with thermal ablation [14,15], to name a few.

Although the prognostication of radiomics has been investigated in lung cancer patients treated with SBRT [16], little is known about the role of radiomics in predicting the clinical outcomes of patients with pulmonary metastases treated with SBRT. With this study, we sought to investigate noninvasive pretreatment CT radiomic features of pulmonary metastases and their peri-tumoural regions that could predict recurrence-free survival (RFS) and overall survival (OS) post SBRT.

To quantify tumour heterogeneity, first-order radiomic features related to CT density histograms and statistical metrics were extracted using two distinct methods: (1) a data-driven technique based on functional principal components analysis (FPCA) using the segmented 3D tumour and peri-tumoural regions [17,18], which extracts few independent features that best characterize the variability of the CT histograms; and (2) 2D CT texture analysis (CTTA) using a filtration-histogram and statistical-based approach that extracts and enhances features of different sizes and intensity variations from CT using the commercially available software Tex-RAD [6]. This study explores the relative values and limitations of these two methods.

## Material and methods

### Patients and metastases

The study was approved by the Research Ethics Board of Sunnybrook Health Science Centre, and patient consent was waived because of the retrospective nature of the study (project

identification number: 362–2017). All methods were performed in accordance with the relevant guidelines and regulations.

One hundred eleven patients underwent SBRT of 163 lung metastases from November 2010 until March 2017. Data were accessed for research purposes between December 2019 and December 2020. Only the study's principal investigator, who was one of the authors, had access to information that could identify individual participants during the data collection. No other authors could identify individual participants. Fifty-seven patients were female and 54 patients were male, with an average age of 67 years (range: 34–90). Ninety-one patients had one pulmonary metastasis, 16 patients had 2 pulmonary metastases and 2 patients had 3 and 4 metastases, respectively. Eighty-eight metastases were from colorectal cancer; 25 metastases were from lung cancer; 18 metastases were from renal cell cancer; 16 metastases were from breast cancer; 6 metastases were from uterine cancer; 3 metastases were from melanoma and bladder cancer, respectively; 1 metastasis was from head and neck and esophageal cancer, respectively, and 1 metastasis was from an unknown primary (S1 Table in S1 File).

## SBRT technique

The SBRT technique is described in the supplementary material.

## Chest CT technique

All patients underwent a multidetector chest CT before the SBRT and within 1 month of the SBRT start date. Most of the studies were conducted in our institution using GE LightSpeed Plus or LightSpeed VCT 64 multidetector CT scanners. CTs were acquired volumetrically post contrast medium administration (2 ml/kg body weight of non-ionic contrast medium) at a flow rate of 3.5 ml/s. Technical parameters were as follows: tube voltage 120 kVp, beam pitch 0.984:1, section collimation 64 x 0.625, and image reconstruction thickness 2.5 mm. One hundred and one patients were scanned with a contrast medium, and 10 patients underwent an unenhanced study.

## Follow-up and evaluation of patient clinical outcomes

Patients were followed-up with CT of the chest and abdomen every 4 months for the first 3 years after SBRT and every 6 months thereafter. Of note, in our institution, any lung SBRT should not be delivered concurrently with adjuvant chemotherapy.

RFS was measured from the initiation of SBRT to the earliest of recurrence (local progression or new metastases), death or final follow-up visits for patients who remained alive. OS was defined as the time from the SBRT start date until death or final follow-up visit for patients who remained alive [19,20]. Local recurrence was defined as a metastasis relapsing within or ≤1 cm beyond the planning target volume, and with consecutive enlargement of the metastasis seen on 2–3 CT scans [21]. Control of any recurrence (AR) was defined as the absence of any recurrence of all types (local, lobar, regional, or distant).

Local recurrence was assessed for each pulmonary metastasis treated. RFS, AR and OS were calculated based on each patient treated. When the patient had more than one metastasis, only the largest lesion was used for the lesion-specific geometric, CT density-based empirical or texture- (filtration-histogram and statistical-) based radiomic features.

## Clinical indication for SBRT

Indications for SBRT were as follows: (1) single metastasis and (2) oligometastases, for which the goal was to irradiate all sites of disease (≤5 active metastases); (3) oligoprogression, for

which the goal was to irradiate only those tumours (≤5) that were progressing while a systemic therapy strategy was controlling all other sites of disease; and (4) dominant areas of progression, for which the goal was to irradiate dominant tumours, even if other tumours were progressing, usually in patients with indolent disease.

## Radiomic feature extraction

Two distinct methods were used to extract the radiomic features:

### 1. Data-driven 3D CT histogram-based features extraction

This is a data-driven method based on FPCA of the CT density histogram of the whole segmented metastasis or the corresponding peri-tumoural region. The FPCA method has been previously described in the context of lung tumour classification [20,22]. It extracts the few independent main modes of variation of tumour and peri-tumour CT density histograms in the patient cohort without using predefined statistics. FPCA modes of variation are used for CT density data exploration of the study dataset and for scoring each smooth CT density histogram according to its profile. The scores are then used as imaging features in survival analysis or prognostic models.

The process of data-driven 3D histogram-based features included the following steps: lung metastases were segmented via the commercial software Vitrea software v.7.6 (Canon Medical systems, Otawara, Japan) by using the semi-automatic contouring tool on multiplanar reformatted images with interpolation among non-contiguous slices and the use of manual correction whenever necessary. For each metastasis, linear measurements, volume, mean CT density (attenuation) and its standard deviation were automatically computed. For each segmented metastasis, a peri-tumoural region was automatically extracted as a 3 mm thick region surrounding the metastasis. The peri-tumoural region was not manually edited and could include some tissue outside the lung parenchyma in case of juxtapleural metastasis. Histograms with CT densities for each metastasis and each peri-tumoural region were exported as separate.csv files for further analysis.

Each CT density histogram was converted in smooth curves defined between -1000 HU and 500 HU using Ramsey's method for frequency distributions [23] as previously described [18]. FPCA was applied separately for the metastasis histograms and for the peri-tumoural histograms to extract the main modes of variations of the CT attenuation curves. This FPCA is based on Petersen and Müller's FPC method for frequency distributions using the R-library "fdadensity"[17], which has been previously described [18,20]. The resulting functional principal components (FPCs) were used to interpret the variation among CT histograms. The first three prominent FPCs were considered in this study, excluding higher rank FPCs after reaching the threshold of <10% of the fraction of variance explained (FVE) for the third FPC (F3 or Peri-F3). The associated FPC scores for the metastases and for the peri-tumoural regions were added to the list of predictors (including linear measurements and demographic, texture-based and density-based features) of a patient's recurrence-free survival. A comprehensive description of the FPCA method used in the current study is available in the Supplementary Materials Section: Functional Principal Component Analysis.

### 2. Predefined 2D filtration histogram and statistical-based CT texture analysis (CTTA)

CTTA is a 2D-image based filtration-histogram and statistical-based texture analysis method. The lesions were segmented with a 2D region of interest (ROI) using a free-hand applied contour at the slice with the largest cross-sectional dimensions of the tumour and manually corrected if needed. The process of extraction of the radiomic features comprises an

initial filtration step using a Laplacian of Gaussian (a band-pass filter similar to a non-orthogonal Wavelet), which extracts and enhances features of different sizes and gray levels or intensity variations corresponding to the spatial scale filter (SSF) in radius. SSFs varied from 0 (without filtration; a conventional image), 2mm (fine-texture scale), 3mm (medium-texture scale), 4mm (medium-texture scale), and 5mm (medium-texture scale) to 6mm (coarse-texture scale) [24]. In total, there were 36 filtration-histogram- and statistical-based texture features, comprising 6 texture metrics x 6 SSFs extracted from the tumoural ROI using the commercially available research software TexRAD (Feedback Medical Ltd., UK). This widely used approach can be done with 2D ROIs and has been described in numerous oncologic studies [6,8,12,25,26].

## Statistical analysis

**The CT density histograms.** Demographic, CT acquisition and imaging features were summarized for each RFS group (RFS = 0, RFS = 1) and OS group (OS = 0, OS = 1), with median, inter-quartile range and Mann-Whitney tests (continuous variables) or Chi-squared tests (categorical variables; Tables 1 and 2).

**Correlation analysis.** Correlations between demographic, geometric, texture-based and CT density-based continuous variables were explored using correlograms with Rho Spearman rank correlation to reduce the number of variables in the multivariate models while limiting multicollinearity issues. The linear regressions between entropy and metastasis size or ROI area showed clear breakpoints, so these were evaluated using Muggeo's method for segmented regression with pseudo-scores [27].

**Survival analysis.** Kaplan-Meier plots with log-ranks tests were used to assess the association between clinical or radiomic features and RFS, AR and OS. Continuous variables were dichotomized using below and above median groups. Median survival time could not be computed for all groups when RFS or OS did not reach 50% at the end of the study follow-up. Consequently, the restricted mean survival time (RMST) [28] was computed for each group of significant variables. RMST represents the average event-free survival time up to a pre-specified time point (here: 3 years or 36 months). It is equivalent to the area under the Kaplan-Meier curve from the beginning of the study through that time point. The inter-group RMST difference for RFS indicates the gain or loss in the recurrence-free survival time between groups 1 and 2 during this period (36 months).

Our original variable list represents a medium dimensional variable set, too large for the usual stepwise variable selection. Consequently, as the first step of the variable selection, we applied a model-based boosting technique to screen the potential best predictors of RFS, OS, and AR. Statistical boosting provides intrinsic variable selection eliminating non-significant variables with the computation of multivariable Cox regression models. In this study, we used the restricted likelihood-based boosting algorithm adapted for Cox regression models of the "mboost" R library [29]. A simple number of 100 iterations was determined during pilot tests as adequate to select the relevant predictor candidates for the final models. Variable importance in the model-based boosting framework was computed to rank the candidate Cox model predictors. (See S5 and S6 Figs for a list of preselected variables including the clinical variables "clinical indication" and "lung primary cancer origin").

In the second step, multivariate Cox model building was performed with the preselected variables, excluding highly correlated secondary variables (such as peri-F1) and using a simple backward variable selection (entering a variable if P < 0.1 and removing the variable if P > 0.15). The analysis for the proportional hazard assumption of the Cox models RFS-1 and RFS-2 showed some time-varying effect for the variable age. Consequently, these models were

**Table 1. Demographic, CT-Acquisition parameters and potential RFS predictors in the two Groups of RFS: 0 = recurrence free survival, 1 = recurrence or death.**

| Variable | RFS:0 [recurrence free survival] N = 28 | | RFS:1 [recurrence or death] N = 83 | | |
|---|---|---|---|---|---|
| **Demographic and CT Acquisition (discrete variables)** | | | | | |
| Sex | F: 17 (61%). M:11 (39%) | | F: 40 (48%) M: 43 (52%) | | P: 0.2538 |
| Contrast CT: C/NC | C: 25 (89%). NC: 3 (11%) | | C: 76 (92%). NC: 7 (8%) | | P: 0.7167 |
| **Additional Clinical Variables (Primary cancer origin and clinical indication)** | | | | | |
| Colorectal: Yes/No | Yes: 16 (57%). No: 12 (43%) | | Yes: 37 (45%). No: 46 (55%) | | P: 0.2519 |
| Primary Cancer Origin | Lung: 2 (7.1%). Breast: 3 (10.7%). Colorectal: 16 (57.1%). Renal cell carcinoma: 4 (14.3%). Melanoma: 0 (0%). Other: 3 (10.7%) | | lung: 18 (21.7%). Breast: 9 (10.8%). Colorectal: 37 (44.6%). Renal cell carcinoma: 9 (10.8%). Melanoma: 2 (2.4%). Other: 8 (9.6%) | | P: 0.5406 |
| Primary Cancer origin: Lungs vs. other | Lung: 2 (7.1%). Other: 26 (92.9%) | | Lung: 18 (21.7%). Other: 65 (78.3%) | | P: 0.0848 |
| Clinical Indication -Metastatic Status | 1. Single met or Oligomet: 23 (82.1%) 2. Oligoprogression or dominant areas of progression: 5 (17.9%) | | 1. Single met or Oligomet: 49 (59.0%) 2. Oligoprogression or dominant areas of progression: 34 (41.0%) | | **P: 0.0275** |
| **Age, linear measurements and volume** | | | | | |
| | Median | 25–75 Perc. | Median | 25–75 Perc. | P-value (Mann-Whitney) |
| Age | 69.5 | 58.5 78 | 66 | 59 75 | 0.5591 |
| Size | 1.45 | 1.1 2.6 | 1.9 | 1.30 2.80 | 0.1237 |
| Volume ml | 3111.9 | 1121 8721 | 4109.5 | 1584 8294 | 0.5593 |
| Mean diam. | 17.55 | 12.15 26.2 | 19.7 | 14.33 26.75 | 0.3366 |
| Max diam. | 20.55 | 14.75 32.0 | 22.5 | 17.90 31.75 | 0.3683 |
| **CT density histogram features (Vitrea)** | | | | | |
| Min HU | -920.5 | -1008–837 | -924 | -975–853 | 0.9376 |
| Max HU | 283.5 | 172 442 | 300 | 216 412 | 0.659 |
| F1 | 0.173 | -0.192 0.449 | -0.0121 | -0.367 0.335 | 0.0834 |
| F2 | 0.0305 | -0.139 0.209 | 0.00581 | -0.207 0.185 | 0.4429 |
| F3 | 0.0266 | -0.061 0.130 | 0.00226 | -0.091 0.095 | 0.3998 |
| **Peri-F1** | -0.189 | -0.458 0.131 | -0.0139 | -0.290 0.397 | **0.0505** |
| Peri-F2 | -0.0318 | -0.122 0.161 | 0.0277 | -0.135 0.194 | 0.5233 |
| Peri-F3 | 0.0168 | -0.109 0.079 | 0.00619 | -0.081 0.120 | 0.3923 |
| **Filtration-histogram and statistical based CT texture analysis (CTTA) features (TexRAD)** | | | | | |
| ssf0-mpp | 66.80 | 50.89 87.02 | 72.55 | 51.87 93.30 | 0.6763 |
| ssf4-mpp | 44.895 | 32.55 61.71 | 55.2 | 39.53 66.89 | 0.1335 |
| ssf0-mean | 52.41 | 32.647 76.760 | 59.12 | 39.47 78.10 | 0.5145 |
| ssf4-mean | 1.2 | -6.63 4.44 | 2.08 | -6.01 9.69 | 0.5277 |
| ssf0-sd | 52.21 | 42.71 68.00 | 57.14 | 43.72 64.20 | 0.6346 |
| ssf4-sd | 59.27 | 42.67 70.00 | 64.77 | 49.04 81.29 | 0.1639 |
| ssf0-skewness | 0.51 | 0.17 0.75 | 0.36 | 0.11 0.70 | 0.5277 |
| ssf4-skewness | 0.095 | -0.39 0.47 | 0.03 | -0.41 0.35 | 0.4113 |
| ssf0-kurtosis | 0.25 | -0.25 1.07 | 0.40 | -0.26 1.30 | 0.8947 |
| ssf4-kurtosis | -0.35 | -0.81 1.05 | -0.61 | -0.91 0.15 | 0.1053 |
| **ssf0-entropy** | 4.9 | 4.54 5.10 | 4.71 | 4.37 4.92 | **0.0521** |
| **ssf4-entropy** | 4.82 | 4.35 5.07 | 5.08 | 4.61 5.35 | **0.0353** |

P-values for categorical variables result from chi2 tests. Bold: Significant (P<0.05) or marginally significant (P = 0.05) P-values.

**Table 2. Demographic, CT-Acquisition parameters, and potential Overall Survival (OS) predictors in the two Groups of OS: 0 = Survival, 1 = Death.**

| Variable | OS:0 (0 = survival) N = 74 | | OS:1 (1 = death) N = 37 | | |
|---|---|---|---|---|---|
| **Demographic and CT Acquisition (discrete variables)** | | | | | |
| Sex | F: 39 (53%) M: 35 (47%) | | F: 18 (49%). M: 19 (51%) | | P: 0.688 (Chi2 test) |
| Contrast/NoContrast CT (C/NC) | C: 66 (92%). NC: 7 (8%) | | C: 35 (95%). NC: 3 (5%) | | P: 0.351 |
| **Additional clinical variables (Primary cancer origin and clinical indication)** | | | | | |
| Colorectal: Yes/No | Yes: 39 (53%). No: 35 (47%) | | Yes: 14 (38%). No: 23 (62%) | | P: 0.141 |
| Primary Cancer origin | Lung: 8 (10.8%). Breast: 10 (13.5%). Colorectal: 39 (52.7%). Renal cell carcinoma: 11 (14.9%). Melanoma: 0 (0%). Other: 6 (8.1%) | | Lung: 12 (32.4%). Breast: 2 (5.4%). Colorectal: 14 (37.9%). Renal cell carcinoma: 2 (5.4%). Melanoma: 2 (5.4%). Other: 5 (13.5%) | | P: **0.0079** |
| Primary Cancer origin: Lung vs. other | Lung: 8 (10.8%). Other: 66 (89.2%) | | Lung: 12 (32.4%). Other: 25 (67.6%) | | P: **0.0054** |
| Clinical Indication -Metastatic Status | 1. Single met or Oligomet: 49 (66.2%) 2. Oligoprogression or dominant areas of progression: 25 (33.8%) | | 1. Single met or Oligomet: 23 (62.2%) 2. Oligoprogression or dominant areas of progression:14 (37.8%) | | P: 0.675 |
| **Age, linear measurements, and volume** | | | | | |
| | **Median** | **25–75 Perc.** | **Median** | **25–75 Perc.** | P-value (Mann-Whitney) |
| Age | 67 | 59 78 | 66 | 58.8 73.3 | 0.612 |
| **Size** | 1.55 | 1.00 2.60 | 2.1 | 1.7 3.5 | **0.0036** |
| Volume ml | 3456.9 | 1289 6775 | 5103.8 | 2399 12523 | 0.0632 |
| Mean diam. | 19.0 | 12.70 26.00 | 21.9 | 17.13 31.23 | 0.0636 |
| Max diam. | 21.9 | 14.50 30.70 | 23.3 | 18.93 36.83 | 0.0687 |
| **CT Density histogram features (Vitrea)** | | | | | |
| Min HU | -922 | -985–839 | -919 | -976.3–858.8 | 0.8853 |
| Max HU | 286.5 | 200 430 | 298 | 229.5 403 | 0.5798 |
| F1 | 0.1563 | -0.308 0.351 | -0.07964 | -0.467 0.362 | 0.2575 |
| F2 | 0.008041 | -0.192 0.185 | 0.01349 | -0.175 0.202 | 0.9750 |
| F3 | 0.01503 | -0.109 0.108 | 0.01810 | -0.0504 0.0755 | 0.9701 |
| Peri-F1 | -0.09276 | -0.310 0.220 | -0.01553 | -0.359 0.484 | 0.5073 |
| Peri-F2 | 0.001718 | -0.127 0.188 | 0.02774 | -0.138 0.243 | 0.3710 |
| Peri-F3 | 0.01541 | -0.0812 0.118 | -0.01051 | -0.122 0.0837 | 0.4054 |
| **Filtration-histogram and statistical based CT texture analysis (CTTA) features (TexRAD)** | | | | | |
| ssf0-mpp | 68.29 | 49.86 89.86 | 68.53 | 52.92 88.15 | 0.6479 |
| ssf4-mpp | 51.10 | 39.22 63.11 | 54.56 | 36.37 67.54 | 0.6036 |
| ssf0-mean | 52.74 | 35.84 75.92 | 58.45 | 34.76 78.48 | 0.6843 |
| ssf4-mean | 2.61 | -5.26 9.04 | -0.2 | -16.48 7.33 | 0.2334 |
| ssf0-sd | 53.60 | 42.69 65.04 | 51.74 | 43.12 68.44 | 0.9178 |
| ssf4-sd | 63.67 | 44.43 78.17 | 69.07 | 53.14 87.36 | 0.1223 |
| ssf0-skewness | 0.465 | 0.13 0.72 | 0.500 | 0.20 0.76 | 0.6014 |
| **ssf4-skewness** | 0.135 | -0.21 0.44 | -0.160 | -0.64 0.31 | **0.0226** |
| ssf0-kurtosis | 0.27 | -0.32 1.03 | 0.30 | -0.18 1.29 | 0.6389 |
| ssf4-kurtosis | -0.56 | -0.94 0.17 | -0.49 | -0.78 0.89 | 0.3245 |
| **ssf0-entropy** | 4.78 | 4.47 5.00 | 4.95 | 4.73 5.14 | **0.0417** |
| **ssf4-entropy** | 4.92 | 4.55 5.27 | 5.18 | 4.67 5.39 | **0.0443** |

P-values for categorical variables are based on chi2 tests. Bold: significant (P<0.05) P-values.

estimated by stratifying the time in two groups: before and after one year of patient follow-up for the variable age. Five final multivariate Cox models were selected based on their predictive performances (bias corrected C-indexes) and parsimony (fewer than five variables). These

ones were estimated with the rms [30] and survival [31] R-libraries. Their predictive values were computed using a C-index with overfitting correction through resampling (2000 Bootstrap iterations) following the method of Harrell et al. [32]. A comparison using an alternative overfitting correction method (cross-validation) showed extremely similar results for the corrected C-indices.

The proportional hazards assumption of the best model was tested for each predictor using Schoenfeld's partial residual plots [33].

The significance level was set at < 0.05. All statistical analyses were performed with custom R-scripts and MedCalc v.20.

## Results

### CT-density curve analysis

The FPCA resulted in three FPCs explaining 92% and 87% of the variability of the CT density curves for the metastases and the peri-tumoural regions, respectively. The FPC scores (F1, F2, F3, Peri-F1, Peri-F2, and Peri-F3) associated with specific profiles of the CT density curves were added to the list of existing features.

Fig 1 represents the modes of variation of the three FPCs for the metastases tested in the final model selection. Similarly, the modes of variation of the three FPCs for the peri-tumoral regions are presented in S1 Fig.

### CT-Texture Analysis (CTTA)

CTTA comprises a filtration-histogram on the 2D ROI-based technique where the filtration step extracts and enhances features of different intensity variations and sizes corresponding to the SSF, which varies from SSF = 2mm (fine texture) and 3–5mm (medium texture) to 6mm (coarse texture). SSF = 0 corresponds to no filtration (a conventional image). Quantification of texture comprised mean-intensity, standard-deviation, entropy, mean of positive pixels (MPP), skewness and kurtosis. Median values of the key CTTA features in the 2 groups for RFS and OS are described in Tables 1 and 2.

Demographics, CT-acquisition parameters, and the study's RFS predictors are presented for the two RFS groups in Table 1: 1) RFS: 0 = no recurrence or death, 2) RFS: 1 = recurrence or death. Significant differences between the 2 groups are found in the clinical indications for SBRT (P = 0.0275), entropy (ssf4-entropy: P = 0.03, ssf0-entropy: P = 0.05) and the peri-tumoural parameter Peri-F1 (P = 0.05).

The same variables are shown for the two OS groups in Table 2: 1) OS: 0 = survival, 2) OS: 1 = death.

For simplicity, in Tables 1 and 2, CTTA variable results are only shown for no filtering (SSF = 0) and the medium filter scale (SSF = 4), which appear in the final multivariate models.

Table 2 presents the demographic, radiologic linear and CT-density based information for the two OS patient groups. Significant differences between the 2 groups were found for primary lung cancer origin (P = 0.0054), entropy (ssf4-entropy: P = 0.04, ssf0-entropy: P = 0.04), medium-filtered skewness (ssf4-skewness P = 0.023) and tumour size (P = 0.0036).

### Kaplan-Meier analysis

Kaplan-Meier analysis and its log-rank tests were performed for all variables. Each continuous variable was split into 2 groups below and above their median value. Table 3 presents the variables retained in the multivariate final predictive Cox models for either RFS or OS. The restricted mean RFS time was computed for all significant RFS variables and restricted mean

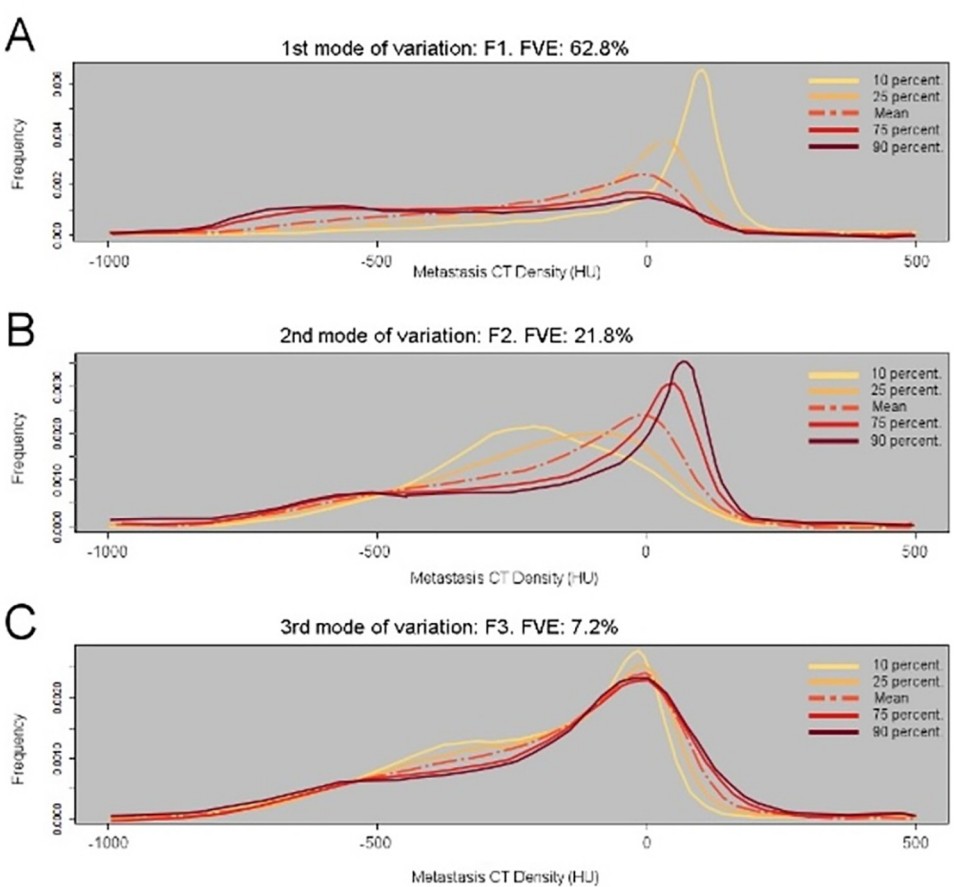

**Fig 1. Functional principal component analysis of the metastasis CT density curves.** A. First mode of variation (F1). B. Second mode of variation (F2). C. Third mode of variation (F3). FVE: Fraction of the variance explained.

survival time was computed for all significant OS variables, with a time horizon = 36 months. See the statistical analysis section for details.

Moreover, four examples of Kaplan-Meier plots with significant differences in RFS (ssf4-entropy and age) and OS (F1 and size) are shown in Fig 2.

The correlation analysis revealed large clusters of highly correlated variables among CTTA-based variables with different filtration scales, as expected. For example, the ssf4-entropy versus ssf6-entropy rank correlation coefficient is 0.97. Moreover, several noteworthy variables show a rank correlation discussed in this study:

- The metastasis density-based F1 vs. the texture-based entropy: ssf4-entropy: **-0.71** 95% CI: [-.79 to -0.60]

- The metastasis density-based F1 versus the tumour volume (log): **-0.61** 95% CI: [-0.71 to -0.47]

- The metastasis texture-based entropy: ssf4-entropy versus metastasis size: **0.73** 95% CI: [0.63–0.81]

- The metastasis texture-based entropy: ssf4-skewness versus metastasis size: **0.11** 95% CI: [-0.08–0.29] NS

**Table 3. Kaplan-Meier Analysis of variables retained in the final multivariate models.**

| Final Variables | Recurrence Free Survival (RFS) P-value (K-M Logrank test) Restricted Mean RFS (months) (95%CI) at 36 months follow-up and p-value for inter-group difference. | Overall Survival (OS) P-value (K-M Logrank test) Restricted Mean OS (months) (95%CI) at 36 months follow-up and p-value for inter-group difference. |
|---|---|---|
| Age (median = 67) | **P = 0.049** Group 1: 13.0 months [9.7 16.3] Group 2: 17.3 months [13.7 20.7] Test RMST: P = 0.0825 (NS) | P: NS |
| Clinical Indication for SRBT– Metastatic status: Group 1. Single met or Oligomet. Group 2. Oligoprogression or dominant areas of progression. | **P = 0.0001** Group 1: 17.8 months [14.8 20.8] Group 2: 9.6 months [6.1 13.1] Test RMST: P = 0.0005 | **NS (P = 0.27)** Group 1: 29.2 months [26.7 31.6] Group 2: 25.3 months [20.9 29.7] Test RMST: P = 0.132 (NS) |
| Primary Cancer–Lung origin Y/N (Y = group2) | **P = 0.0338** Group 1: 16.0 months [13.3 18.7] Group 2: 10.3 months [5.4 15.3] Test RMST: P = 0.05 | **P = 0.0044** Group 1: 48.4 months [26.9 31.6] Group 2: 21.8 months [25.7 30.1] Test RMST: P = 0.0083 |
| Size | P: NS | **P = 0.043** Group 1: 30.1 months [27.3 32.9] Group 2: 25.9 months [22.6 29.2] Test RMST: P = 0.0577 (NS) |
| F1 | P: NS (P = 0.053) | **P = 0.031** Group 1: 25.2 months [21.8 28.7] Group 2: 30.6 months [28.0 33.1] Test RMST: P = 0.0154 |
| ssf4-entropy | **P = 0.026** Group 1: 17.6 months [14.1 21.1] Group2: 12.2 months [9.1 15.4] Test RMST: P = 0.0262 | P: NS |
| ssf4-skewness | P: NS | **P = 0.047** Group 1: 25.9 months [22.7 29.1] Group2: 29.9 months [26.9 32.9] Test RMST: P = 0.0702 (NS) |
| ssf0-kurtosis | | **P = 0.62** Group 1: 28.2 months [25.3 31.2] Group2: 27.7 months [24.4 30.9] Test RMST: P = 0.797 (NS) |
| F2 | P: NS | P: NS |
| Peri-F3 | P: NS | P: NS |

Patient groups for continuous variables are defined as below the median (group 1) and above the median value (group2). Restricted Mean survival time (for RFS) and Restricted Mean Survival Time (for OS) and 95% confidence intervals are indicated for each group with the p-value of the test for RMST difference. NS: Not Significant.

## Association between entropy and metastasis size or ROI area

Dercle et al. [34] reported a correlation between entropy and tumour size and a problematic bias related to small tumour size (below 200 pixels for the 2D ROI). Consequently, we investigated the association between these two variables. The rank correlation between the predictor ssf4-entropy and the tumour (metastasis) size was 0.73 95% CI: [0.63–0.81]. Moreover, a linear model with segmented regression between ssf4-entropy and size revealed a significant breakpoint at 1.48 cm 95% CI: [1.15–1.81], p = 0.0027 (pseudo-score test; Fig 3A). The same method used with ssf0-entropy and log10 2D ROI area (number of pixels) showed a similar significant breakpoint for an area <120 pixels 95% CI: [89–162], p < 0.0001, largely confirming the results of Dercle et al. (Figs 3B and 4).

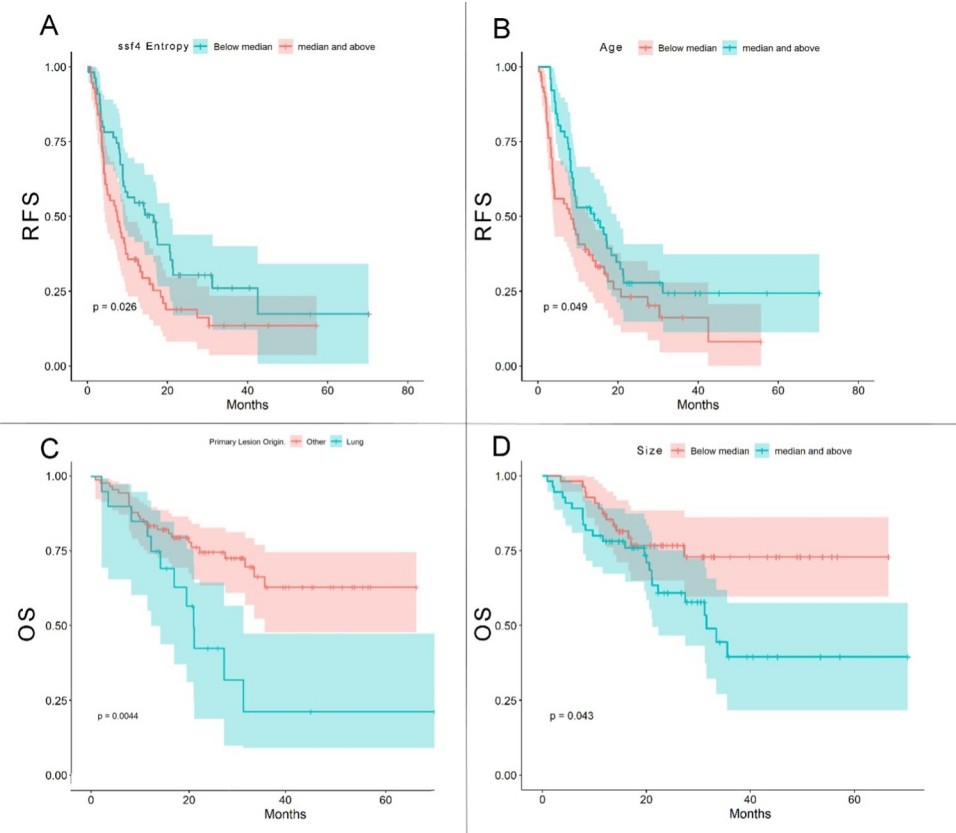

**Fig 2. Kaplan-Meier plots.** A: RFS vs. ssf4-entropy. B: RFS vs. age. C. OS vs. primary lesion origin: Lung. D. OS vs. size.

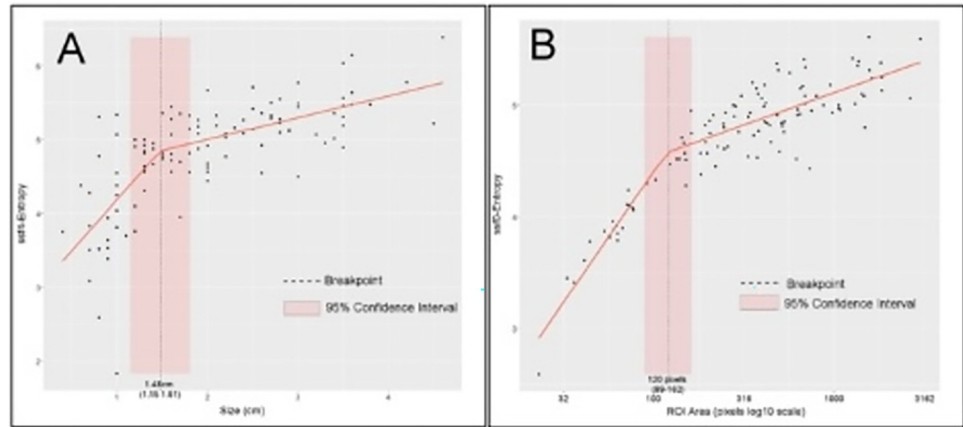

**Fig 3.** A) ssf4-entropy vs. tumour size (cm). The ssf4-entropy is correlated with the tumour size and shows a breakpoint for a size = 1.48 cm 95% CI: (1.15–1.81). B) ssf0-entropy vs. log.10 (area) expressed as the number of pixels in the tumour 2D ROI. A breakpoint is detected for the number of pixels <120 pixels 95% CI:(89–161). Red line: Segmented regression mean value. Overlay: 95% confidence intervals for the breakpoints.

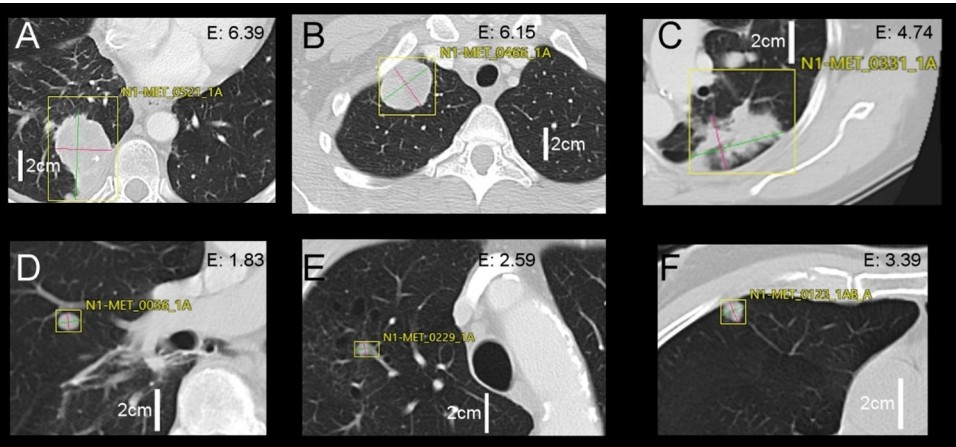

**Fig 4.** A-B-C: Highest medium-filtered entropy metastases. D-E-F: Lowest-entropy metastases. "E" at the top of each image represents the value of ssf4-entropy (Vitrea software v.7.6, Canon Medical systems, Otawara, Japan).

## Effect of the tumour size on CT density-based features F1, F2, F3

The F1 feature was correlated with tumour volume (log.10 scale); Spearman rho -0.60 95% CI [-0.71–.47]. The Bland-Altman plot in Fig 5 shows the agreement of F1 values before and after random removal of 90% of the metastasis's voxels, keeping only 1/10 of the original voxels. The 95% range for the limits of agreement is 0.032, which is 4.7% of the F1 inter-quartile range. In other words, F1 remains remarkably insensitive to the simulated small tumour

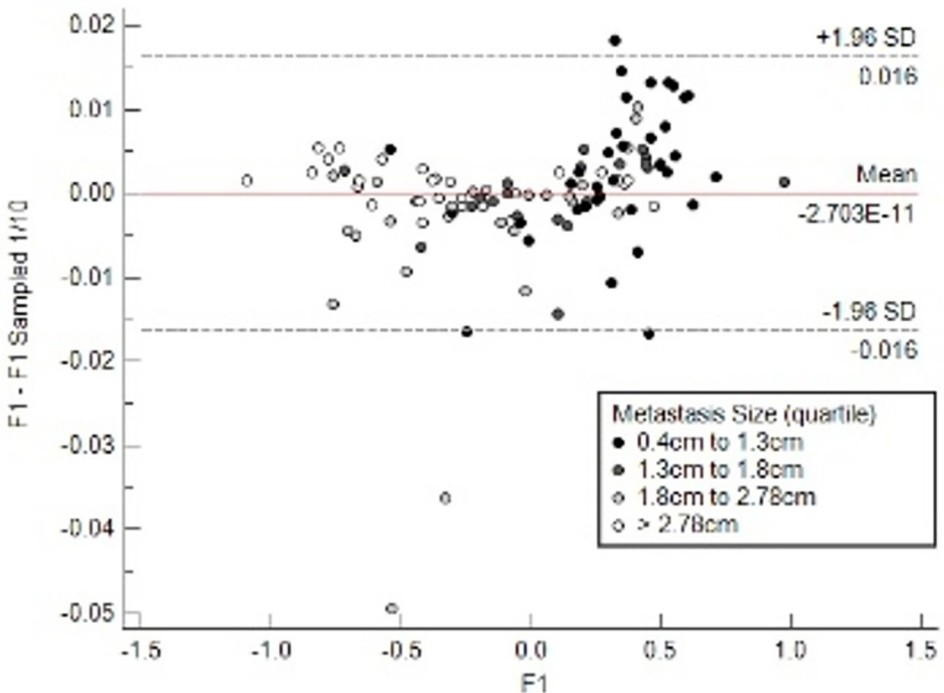

**Fig 5. Bland-Altman plot.** F1 values versus F1 with 1/10 sampled CT histograms. Keeping only 1/10 voxels in the 3D ROI hardly affects the original F1 values (95% limits of agreement: +/- 0.016).

volume even in the smaller size group (0.4 cm to 1.3 cm). F2 and F3 features were similarly unaffected by the simulated volume reduction.

## Multivariate Cox models

Multivariate Cox models were built for each outcome (RFS, OS and AR) with a 2-step variable selection described in the method section. Five final concise multivariate Cox models (with fewer than 5 predictors) were selected based on their highest C-index corrected for overfitting.

Multivariate Cox models are presented in 2 categories: 1. Cox models without clinical variables and 2. Cox models with the following two clinical variables: a) primary cancer origin (1. lung origin versus 2. other non-lung origin) and b) clinical indications for SRBT: metastatic status (1. single metastasis or oligometastases versus 2. oligoprogression or dominant areas of progression).

**Models for Recurrence Free Survival without clinical variables: RFS-1 and RFS-2.** The best Cox model of the first category (RFS-1) for the RFS prediction combined the density-based variables for the metastasis: F1, F2, Peri-F3 (the latter for the peri-tumoural region) and age (C-index 0.62 [0.57–0.69]; Table 4).

In the RFS-1 model, the higher CT density-based predictor F1 was significantly associated with a better RFS [HR: 0.54 (0.38–0.76), P < 0.0001 (other covariates adjusted)].

Similarly, the higher CT density-based predictor F2 was associated with a better RFS [HR: 0.71 (0.51– .99)–P = 0.049]. Moreover, the peri-tumoural predictor peri-F3 was associated with a worse RFS [HR: 1.32 (1.0–1.74) P = 0.05]).

Considering the stratified predictor age, during the first year (<365 days), older age (>67 years) was associated with a better RFS [HR: 0.59 (0.40–0.85) P = 0.0005]. After 1 year, however, the effect of age was no longer significant [HR: 1.19 (95% CI: [0.61–2.33], P = 0.72].

A similar Cox model (RFS-2) was defined with the following variables: ssf4-Entropy, F2 and age with performances similar to the previous one (C-index: 0.62 [0.57–0.67]).

In the RFS-2 model, higher CTTA texture-based predictor ssf4-entropy was significantly associated with a worse RFS [HR: 1.54 (1.16–2.05), P < 0.0026].

The higher CT density-based predictor F2 was associated with a better RFS [HR: 0.71 (0.51–0.99), P = 0.026]. Older age (>67 years) was associated with better RFS [HR: 0.59 (0.40–0.85), P = 0.0005] during the first follow-up year (<365 days). After 1 year, the effect of age was not significant [HR: 1.19 (95% CI: [0.61–2.33], P = 0.72].

**Model for Recurrence Free Survival with clinical variables: RFS-3.** The best predictive Cox model for RFS combined with clinical variables (RFS-3) includes the following predictors: F1, age, clinical indication and lung primary cancer origin (C-Index: 0.67 [0.62–0.72]). See Table 4 and nomogram S2 Fig.

In the RFS-3 model, the higher CT density-based predictor F1 was significantly associated with a better RFS [HR: 0.57 (0.40–0.82; other covariates adjusted), P = 0.0026]. Regarding the clinical indication, patients in group 2 (oligoprogression or dominant areas of progression) had a worse RFS compared to patients in group 1 (single metastasis or oligometastases) [HR: 1.78 (1.10–2.88), P = 0.0189].

Finally, individuals whose primary cancer was of lung origin had a worse RFS compared to those with a primary cancer of non-lung origin [HR: 1.83 (1.06–3.15), P = 0.0293].

**Model for any recurrence with clinical variables: AR-1.** Similarly, a single best Cox model (AR-1) was selected for the prediction of "any recurrence" time to an event, combining the same variables as RFS-3: F1, age, clinical indication and lung primary cancer origin, resulting in similar hazard ratios (C-Index: 0.66 [0.61–0.71]; Table 4).

**Table 4. Hazard ratios for multivariate model predictors of RFS (models RFS-1 RFS-2 RFS-3), any recurrence (model AR-1) and OS (OS-1).**

| Predictors | Hazard Ratios (95% CI) Interquartile | P-Value |
|---|---|---|
| *Radiologic models (CT Density and texture analysis predictors and age)* | | |
| *Cox model RFS-1: RFS ~ F1 + F2 + Peri-F3 + Age (two strata)–C-index: 0.62 [0.57 0.67]* | | |
| F1 | 0.54 (0.38 0.76) | **<0.0001** |
| F2 | 0.71 (0.51 0.99) | **0.049** |
| Peri-F3 | 1.32 (1.00 1.74) | **0.052** |
| Age: follow-up < 1 year | 0.59 (0.40 0.85) | **0.0005** |
| Age: Follow-up ≥ 1 year | 1.19 (0.61 2.33) | 0.720 |
| *Cox model RFS-2: RFS ~ ssf4-Entropy + F2 + Age (two strata)–C-index: 0.62 [0.57 0.67]* | | |
| Entropy (ssf4) | 1.54 (1.16 2.05) | **0.0026** |
| F2 | 0.71 (0.51 0.99) | **0.026** |
| Age: < 1-year follow-up | 0.59 (0.40 0.85) | **0.0005** |
| Age: ≥ 1-year Follow-up | 1.19 (0.61 2.33) | 0.720 |
| **Predictors** | **Hazard Ratios (95% CI) Interquartile** | **P-Value** |
| *Full models with Clinical Indication and Lung—Primary cancer origin* | | |
| *Cox model RFS-3: RFS ~ F1 + Age +Clinical.Indication + Lung.primary.origin. C-index: 0.67 [0.62 0.72]* | | |
| F1 | 0.57 (0.40 0.82) | **0.0026** |
| Age | 0.73 (0.53 1.02) | **0.0616** |
| Clinical.Indication.Bin: group 2 (oligoprogression or dominant areas of progression) vs. group 1 (single met or oligomet) | 1.78 (1.10 2.88) | **0.0189** |
| Lung.primary.origin: Yes | 1.83 (1.06 3.15) | **0.0293** |
| *Cox model AR1: AnyRec ~ F1+Age +Clinical.Indication + Lung.prim.origin. C-index: 0.66 [0.61 0.71]* | | |
| F1 | 0.57 (0.39 0.84) | **0.0037** |
| Age | 0.70 (0.50 0.98) | **0.0386** |
| Clinical.Indication.Bin: group 2 vs. group 1 | 1.79 (1.09 2.94) | **0.0220** |
| Lung.primary.origin: Yes | 1.74 (0.98 3.09) | 0.0574 |
| *Cox model OS-1: OS ~ Size + Skewness (ssf4) + Kurtosis (ssf0) + Lung.prim.origin. C-Index: 0.67 [0.61 0.74]* | | |
| Size | 1.96 (1.18 3.23) | **0.0089** |
| Skewness (ssf4) | 0.65 (0.47 0.88) | **0.0058** |
| Kurtosis (ssf0) | 1.05 (1.02 1.09) | **0.0040** |
| Lung.primary.origin: Yes | 2.91 (1.43 5.90) | **0.0031** |

**Model for overall survival with clinical variables: OS-1.** The single best Cox model (OS-1) was selected for the prediction of "overall survival," combining the variables size, CT texture-based skewness (ssf4), kurtosis (ssf0) and lung primary cancer origin (C-Index: 0.67 [0.61–0.74]). See Table 4 and nomogram S3 Fig.

In the OS-1 model, a larger lesion size was significantly associated with a worse OS [HR: 1.96 (1.18–3.23), P = 0.0089 (other covariates adjusted)]. Higher CTTA skewness (ssf4) was significantly associated with a better OS [HR: 0.65 (0.47–0.88), P = 0.0058]. Higher CTTA kurtosis (ssf0) was significantly associated with a worse OS [HR: 1.05 (1.02–1.09), P = 0.0040]. Finally, patients whose primary cancer was of lung origin had a worse OS compared to those with a primary cancer from a non-lung origin [HR: 2.91 (1.43–5.90), P = 0.0031].

## Discussion

By using two distinct approaches—FPCA (data driven) and CTTA (filtration-histogram and statistics-based)—for evaluating the CT heterogeneity of pulmonary metastases, this study shows that a combination of tumoural and peri-tumoural radiomic features with clinical variables can predict RFS and OS based on pre-SBRT CT images in patients with pulmonary metastases. The clinical variables "clinical indication" and "lung primary cancer origin" are strongly associated with RFS and OS and significantly improve the predictive performances of the Cox models.

### Clinical variables

Older age was significantly associated with longer RFS in Kaplan-Meier (K-M) analysis log-rank tests, showing a restricted mean RFS (time horizon: 3 years) of 17.3 months for the older patient group (≥67 years) versus 13.0 months for the younger patient group (<67 years). Moreover, when considering the first year of study follow-up in the multivariate Cox models RFS-1 and RFS-2, older patients had better RFS than younger patients. In those two models, the age effect was found to be slightly time dependent and not significant after 1 year of follow-up. Age also has a significant effect on AR when combined with F1, clinical indication and lung primary cancer origin (Cox model AR-1). A lower risk of recurrence in the older patient group (patient > 65 years) has also been previously reported in the context of surgical pulmonary resection of colorectal metastases [35] and for OS after pulmonary metastasectomy from colorectal cancer [36]. Similarly, worse recurrence or survival in young patients has been reported in past studies in breast cancer or lung cancer. For example, Sacher at al. noted: "The survival of young patients with NCSLC is unexpectedly poor compared with other age groups, suggesting more aggressive disease biology"[37]. They also found that younger age was associated with an increased frequency of a targetable genotype alteration.

The clinical indication of the metastatic status of SBRT was a major clinical predictor of RFS in our study. Multivariate Cox hazards analysis showed that the group of oligometastases had significantly better RFS or AR compared to the group with oligoprogression and dominant areas of progression. A clinical indication was associated with RFS in K-M analysis (P = 0.0001) with a restricted mean RFS time of 17.8 months for the oligometastases group versus 9.6 months for the oligoprogression and dominant areas of the progression group. In the multivariate Cox model for RFS combined with clinical variables (RFS-3), the clinical indication remained significant. Similarly, the clinical indication was also significant for AR.

Other studies have also shown that oligometastases treated with SBRT had the best outcomes for OS compared to oligoprogression and dominant areas of progression [21,38–40]. Oligoprogression and dominant areas of progression are increasingly in-demand SBRT indications; however, fewer results have been reported. The rationale for these clinical indications is beyond the goal of cure as it is for single metastasis or oligometastases. The intent is rather to delay the need to start or change systemic therapy, potentially improving quality of life, especially when systemic therapy is more toxic [39].

Lung primary cancer origin was a significant predictor of RFS in K-M analysis with a restricted mean RFS time of 10.3 months for lung primary cancer origin versus 16 months for non-lung origin. Lung origin was also found to be a dismal predictor of OS with an RMST survival time of 21.8 months for the lung origin group versus 48.4 months for the non-lung group. A lower OS with the lungs as a primary cancer origin has been found in oligometastatic tumours treated with stereotactic ablative body radiotherapy (SABR) in a study by Chalkidou et al. [41]. Poon et al. [19] reported significant differences in survival curves of various primary origins in which the lung origin group showed worse survival than the colorectal origin group.

Yamamoto et al. reported that an esophageal origin had a worse outcome compared to a colo-rectal origin for OS [42].The lung primary cancer origin in our study remained a significant predictor of both RFS and OS in multivariate Cox models while adjusted for the other covariates.

A higher local recurrence rate has been reported in pulmonary metastases of colorectal origin treated with SBRT [39,43]. Surprisingly, no association between a colorectal origin of pulmonary metastases and RFS was found in our study (P = 0.14). No significant results were found for local recurrence because the number of events of local recurrence was small.

## Radiomic features

**CT-density & CT texture features.** Entropy with or without filtration and the peri-tumoural CT-density-based peri-F1 showed a significant association with RFS on univariate analysis. Moreover, entropy, skewness, mean HU, SD, MPP, the CT density-based biomarker F1, and lesion size showed significant associations with OS.

In the multivariate analysis, the predictive Cox model (RFS-3) combining the data-driven **F1** variable with the clinical variables (clinical indication and lung primary cancer origin) and age, showed the highest predictive value for RFS. Individuals with a high F1 had a significantly better RFS.

Similarly, the Cox regression model without clinical variables (RFS-1) using the data-driven F1, F2 and peri-F3 biomarkers combined with age (time-stratified) showed the highest predictive value for RFS. Individuals with high F1 had a significantly better RFS.

The Cox model without clinical variables (RFS-2) using predefined CT-density-based entropy, data driven F2, and age (time stratified) showed the same predictive value for RFS.

A single predictive multivariate Cox model (OS-1) was retained for the prediction of OS combining lesion size, patient's age, CTTA skewness (ssf4), unfiltered CTTA-Kurtosis (ssf0) and lung primary cancer origin. However, caution about the OS results is needed because of the small number of events (deaths) in the current study (37/111 patients). Size was also found to be a predictor of OS in other studies about SBRT of a single metastasis or oligometastases to the lung [44,45].

Low F1 score values were associated with CT density curves showing a peak of high CT density (0HU to 150HU), whereas high F1 scores were associated with lower risk CT density curves, showing quite a uniform frequency distribution along the CT density range (-1000HU to 500HU) and very few high densities (Fig 1A).

Lower F2 score values showed a concentration of CT densities around -400HU and -100HU (Fig 1B). They were associated with worse RFS in both models, RFS-1 and RFS-2. In other words, independent of the high-density peak previously seen with low F1 scores, an increased tumour density in the -400HU to -100HU range is also associated with earlier recurrence.

The observed (rank) correlation between the F1 score and (log) volume can be attributed to the well-known relationship between intratumour phenotype heterogeneity and tumour growth (S4 Fig) [46].

Data-driven CT density biomarkers have been used in previous studies for lung adenocarci-noma and have been reported to successfully predict the classification of pre-invasive and invasive subsolid nodules [20,22].

Filtered and unfiltered CT texture parameter entropy was significantly associated with RFS with a restricted mean RFS time of 12.2 months for the high entropy group versus 17.6 months for the low entropy group.

In the multivariate model (RFS-2) adjusted for other covariates, ssf4-entropy was found to be the best predefined predictor for RFS. High entropy values were associated with worse RFS.

The correlation of ssf4-entropy with lesion size was noticeable. Size itself was not significantly associated with RFS, whereas it was strongly associated to OS. The predictive value of metastasis size for OS seems well established for liver [47] but remains unclear for pulmonary metastases with reports of significant [44,45,48] and non-significant effects [49].

Entropy was also associated with OS. Entropy was found to be highly correlated with size—the best OS predictor—and therefore was not retained in the final multivariate model, OS-1. Using a K-M analysis of 525 patients. Dercle et al. [34] recently reported a lack of association between entropy in metastases of diverse locations and OS and a substantial effect of filtering. Our results confirm the lack of association of medium- and coarse-filtered entropy with OS but point out the association of non-filtered (ssf0) and low filtered (ssf2) entropy with OS.

CTTA ssf4-skewness was found to be a significant predictor of OS. Unlike entropy, skewness was not correlated with size or the other covariates and thus was retained in the OS-1 Cox model for OS prediction.

The difference in performance between different variables may be related to the radiomic extraction method, or entropy may be better at predicting RFS than OS. For example, whereas data-driven FPCs features were based on the whole 3D CT histogram (from -1000HU to 500HU), the filtration-histogram- and statistics-based CTTA was based on 2D ROIs where pixels with intensity values $\geq$ -50HU were included in the analysis. Regarding 2D- versus 3D-extracted radiomic features, Piazzese et al. reported that 2D features extracted from esophageal cancer patients performed slightly better than 3D ones when evaluated in terms of stability, dimensionality, and intravenous contrast medium dependency [50]. In a study about synchronous metastases, Dercle et al. showed a confounding effect of a small ROI area on the entropy measurement and recommended a minimum ROI area of 200 pixels [34]. In our study, segmented linear regressions with either entropy (ssf4) versus metastasis size (mm) or entropy versus ROI area confirmed significant breakpoints for small ROIs for a size < 1.48cm and entropy (ssf0) for an area < 120 pixels (Fig 2). Because metastases with areas under 200 pixels represent approximately the first small 30% of all cases in our study, it was difficult to apply this cautious inclusion criteria. Nonetheless, the reported performances of medium-filtered entropy with an adjusted hazard ratio of 1.54 (1.16 2.05) in the Cox model RFS-2 and other univariate analysis results confirm its value for predicting RFS in pulmonary metastases. Moreover, the role of entropy as a robust imaging biomarker has been established in previous lung cancer studies. These studies have shown that entropy was associated with tumour metabolism and may allow staging and predict progression free survival, overall survival, and response to treatment [7,9,34,51,52].

## Peri-tumoural density features

Peri-F1 was found significantly associated with RFS with a restricted mean of RFS = 12.9 months for the high peri-F1 group versus 25.2 months for the low peri-F1 group. However, its high correlation with the best metastases' CT features F1 or ss4-entropy excluded it from the multivariate final models. Other studies have also shown the value of peri-tumoural radiomics. Khorrami et al. reported that the combination of peri-tumoural and intra-tumoural radiomic features on baseline CT predicted the response to chemotherapy and was associated with time to progression and overall survival in lung adenocarcinoma [53]. Shan et al. [25], using 2D radiomics analysis on CT enhanced images to predict the early recurrence of hepatocellular carcinoma after resection, reported the increased predictive value of peri-tumoural radiomics models compared to intra-tumoural models.

## Limitations

Limitations of the study include the inability to rule out inherent methodological issues given the retrospective nature of the study. Moreover, previous radiomics-based survival analyses of lung tumours included hundreds to more than one thousand radiomic features. In contrast, the current study focused on quantifying heterogeneity using two radiomics techniques: one using a well-established and published filtration-histogram- and statistics-based CTTA technique and the other using novel data-driven CT density features. The predictive performance of higher order radiomic features was not addressed and deserves a future study with a larger cohort.

The relatively small size of the study cohort prevented us from testing the Cox models in a fully independent validation set. However, both the bootstrap correction for the predictive accuracy (C-Index) of overfitting and the very small number of model predictors suggest a good generalizability of the predictive models.

Moreover, the small sample size did not allow a reliable per-lesion survival analysis of the local recurrences (N = 26) as a specific outcome, despite its valuable clinical information about SBRT metastases treatment. Similarly, the results of the current study apply to lung metastases of heterogeneous origin (lung, colorectal cancer, etc.). Because of sample size limitations, further stratification was not possible. However, no significant interaction term between the radiomic predictor, either entropy or F1, and the variable "lung as primary origin" was found in the multivariate models, suggesting that the effect of the radiomic variable does not depend on the primary tumour origin, at least among the lung versus other groups. Besides RFS and overall survival analysis, the analysis of recurrences alone would deserve a more accurate (but more complex) semi-competing risk approach taking into account the competing risk of death while predicting the recurrence alone [54].

Further studies with enough events per primary cancer origin subgroup will be needed to find out whether the best radiomic predictors and their effects because hazard ratios vary with the origin of the primary cancer. Moreover, bootstrapping of corrected C-indices would be reduced if more parameters were added to the best final Cox models as expected with an efficient overfitting correction.

Fewer than 10% of the cases included in the current study were acquired without intravenous CT contrast, adding potential heterogeneity to the radiomic analyses. However, the adjustment of the Cox models for CT contrast did not affect the model performances. Stability of CT density and texture features relative to the presence or absence of intravenous contrast have been recently reported in esophageal cancer survival analysis [50].

Because of the long period of follow-up in our study, systemic treatment options that could have followed the SBRT might have inevitably changed and evolved since the treatment of the earliest patients, so that the predictive models might not reflect updated clinical protocols. Nevertheless, in this study, the systemic treatments that the patients might have undergone after the use of SBRT were not factored into the analysis. Moreover, SBRT, which was the main treatment for these pulmonary metastases, was used in this cohort for the same 4 clinical indications throughout the study period as described in Methods.

We did not quantify the peri-tumoural radiomics based on the CTTA method. This could be done in future studies because it has been shown previously that CTTA in breast margins could differentiate between DCIS and IC on mammography, whereas MRI texture analysis of edema surrounding brain lesions could differentiate between primary lesions and single brain metastases [55,56].

Finally, the predictive performances of our best models (C-index: 0.66 and 0.67) for RFS and overall survival are in the range of recent similar studies. For the RFS predictions of

patients with colorectal peritoneal metastases undergoing cytoreductive surgery and using clinical variables only, Dietz et al. [57] reported a C-index of 0.64. Sanli et al. [58] reported a predictive survival model with spinal bone metastases with a C-index of 0.669 (0.598–0.740) using composite clinical and radiomic score predictors. Recently, Liao et al. [59] presented a deep learning MRI radiomic model with a C-index of 0.75 to predict survival in patients with brain metastases treated with gamma knife surgery. Based on contrast-enhanced CT for brain metastases, Zhang et al [60] reported a C-index of 0.66 when using both clinical and radiomic predictors. Finally, using contrast-enhanced MRI images, Zhou et al [61] reported a C-index of 0.78 for clinical and volume-based models in patients with brain metastases who were undergoing stereotactic radiosurgery. These performances remain low for definite clinical applicability and should be considered as first steps toward future predictive survival models. Methodologic improvement for better recurrence and predictive survival models are suggested by the recent literature including large dataset prospective studies, homogeneous cancer origins [62], and deep-learning survival models following the predictor selection [59]. Moreover, a one-point radiomic and morphologic analysis ignores most of the dynamic of the cancer progression. Short-term radiomic and morphologic changes may be possible with accurate automatic segmentation, potentially adding predictive power without requiring long follow-up periods. This post-treatment serial follow-up approach may contribute to early recurrence prediction and may be combined with new genetic tests such as serial circulating tumour DNA (ctDNA), recently evaluated for predicting recurrence in colorectal cancer liver metastasis surveillance studies [63].

## Conclusions

Concise predictive models including CT heterogeneity markers of the metastases and their peri-tumoural regions, the patients' age, clinical indication and lung primary cancer origin can help to identify, prior to SBRT treatment, those patients with probable earlier recurrence or a dismal prognosis so that personalized medicine can be applied more aggressively.

Overall, pretreatment prognostication of survival outcomes for pulmonary metastases treated with SBRT remains under investigation, and larger studies are needed in the future with external validation to confirm these findings.

## Supporting information

**S1 Fig. Main modes of variation of the peri-tumoral CT density histograms.** A. First main mode variation Peri-F1 of the peri-tumoral CT density histogram from low density homogeneous distribution (yellow) to heterogeneous bimodal distribution (brown). B. Second mode of variation Peri-F2. C. Third mode of variation Peri-F3. D. Example of peri-tumoral region with low Peri-F1 (top) and high Peri-F1 (bottom) (Vitrea software v.7.6, Canon Medical systems, Otawara, Japan).
(JPG)

**S2 Fig. Nomogram for model RFS-3 (with clinical variables).** Hazard ratios are presented with their 95% confidence intervals.
(JPG)

**S3 Fig. Nomogram for model OS-1 (with clinical variables).** Hazard ratios are presented with their 95% confidence intervals.
(JPG)

**S4 Fig. Scatterplot CT density F1 vs. metastasis volume (log) with linear regression line.** The CT density variable F1 appears fairly correlated with the metastasis volume (log). Linear

correlation r = 0.59.
(JPG)

**S5 Fig. Variable importance for multivariate RFS model variable selection.** Left: Original variable list (including clinical variables). Right: Selected variables for Recurrence Free Survival (RFS) multivariate Cox model building after 100 boosting iterations ranked by decreasing importance (% in-bag reduction risk).
(JPG)

**S6 Fig. Variable importance for multivariate RFS model variable selection.** Left: Original variable list (including clinical variables). Right: Selected variables for Recurrence Free Survival (RFS) multivariate Cox model building after 100 boosting iterations ranked by decreasing importance (% in-bag reduction risk).
(JPG)

**S1 File.**
(DOCX)

## Author Contributions

**Conceptualization:** Pascal Salazar, Anastasia Oikonomou.

**Data curation:** Patrick Cheung, Anastasia Oikonomou.

**Formal analysis:** Pascal Salazar.

**Investigation:** Pascal Salazar, Anastasia Oikonomou.

**Methodology:** Pascal Salazar.

**Resources:** Anastasia Oikonomou.

**Software:** Pascal Salazar, Balaji Ganeshan.

**Supervision:** Anastasia Oikonomou.

**Validation:** Pascal Salazar.

**Writing – original draft:** Pascal Salazar.

**Writing – review & editing:** Pascal Salazar, Patrick Cheung, Balaji Ganeshan, Anastasia Oikonomou.

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
