## [Decision Letter · Decision Letter 0]

9 Jul 2024

PONE-D-24-09740Predefined and data-driven CT radiomics predict recurrence-free and overall survival in patients with pulmonary metastases treated with stereotactic body radiotherapyPLOS ONE

Dear Dr. Oikonomou,

Thank you for submitting your manuscript to PLOS ONE. After careful consideration, we feel that it has merit but does not fully meet PLOS ONE’s publication criteria as it currently stands. Therefore, we invite you to submit a revised version of the manuscript that addresses the points raised during the review process.

We look forward to receiving your revised manuscript.

Kind regards,

Lorenzo Faggioni, M.D., Ph.D.

Academic Editor

PLOS ONE

“I have read the journal's policy and the authors of this manuscript have the following competing interests:

PS is an employee of Canon Medical Systems.

BG (who was not a data controller/processor for this study) is the co-founder/co-inventor of TexRAD texture analysis software used in this study and a shareholder (not an employee) of Feedback Plc., a UK-based company which owns, develops, and markets the TexRAD texture analysis software.

PC declares no competing interests.

AO declares no competing interests.”

Reviewers' comments:

Reviewer's Responses to Questions

**Comments to the Author**

1. Is the manuscript technically sound, and do the data support the conclusions?

Reviewer #1: Yes

2. Has the statistical analysis been performed appropriately and rigorously? 

Reviewer #1: Yes

3. Have the authors made all data underlying the findings in their manuscript fully available?

Reviewer #1: Yes

4. Is the manuscript presented in an intelligible fashion and written in standard English?

Reviewer #1: Yes

5. Review Comments to the Author

Reviewer #1: Thank you for providing me with the opportunity to review this insightful manuscript. While the study contributes to the expanding literature on radiomics in oncology by highlighting its important role, it is crucial to acknowledge several limitations and areas for improvement to enhance the clinical utility and reliability of predictive models within this patient population.

i) Regarding the C-index values obtained in this study, which range from 0.6 to 0.7, I think the authors should thoroughly discuss this point. It would be beneficial to explore potential strategies for enhancing these values.

ii) The authors mentioned "univariate (Kaplan-Meier)" in the abstract and main text. However, Kaplan-Meier is not a comparable method. This might refer to "log rank" or a similar statistical test.

iii) The administration of adjuvant chemotherapy after SBRT significantly affects RFS or OS. Information on adjuvant chemotherapy should be included for a comprehensive understanding of the treatment outcomes.

6. PLOS authors have the option to publish the peer review history of their article (what does this mean?). If published, this will include your full peer review and any attached files.

Reviewer #1: No

---

## [Author Response · Author response to Decision Letter 0]

12 Aug 2024

We would like to thank the reviewers for their meaningful comments which have been taken into account for the manuscript revision:

i) Regarding the C-index values obtained in this study, which range from 0.6 to 0.7, I think the authors should thoroughly discuss this point. It would be beneficial to explore potential strategies for enhancing these values.

Response: 

We fully agree with the reviewer’ comments on the C-index performances. The issue on the limited predictive values of our final models is now briefly discussed and acknowledged at the end of the section “limitations” lines 568-572. 

ii) The authors mentioned "univariate (Kaplan-Meier)" in the abstract and main text. However, Kaplan-Meier is not a comparable method. This might refer to "log rank" or a similar statistical test.

Response: 

We agree with the reviewer’ comments on the status of the Kaplan-Meier analysis. We removed the “univariate” adjective which seems confusing and added the reference of the log-rank tests which are indeed the inference part of the Kaplan Meier analysis (strictly speaking the Kaplan-Meier curves correspond to the non-parametric survival curve estimator) both in the abstract (‘Results’ section) and in the Discussion – ‘Clinical variables’ section – line 406)

iii) The administration of adjuvant chemotherapy after SBRT significantly affects RFS or OS. Information on adjuvant chemotherapy should be included for a comprehensive understanding of the treatment outcomes.

Response: 

We agree with the reviewer’ comments on the need for information about potential adjuvant chemotherapy. 

In our institution, any lung SBRT should NOT be delivered concurrently with adjuvant chemotherapy. A specific mention now appears in the Material and Method section - Follow-up and evaluation of patient clinical outcomes – lines 117-118

Besides these changes: For more study reproducibility, commented R-code scripts with input and output test files corresponding the computation of the Functional Principal Components of the CT histograms for metastases, peri-metastases regions, CT histogram smoothing and others scripts are now available at: https://github.com/psalaz83/Metastases_Radiomics.git See also in the same link the README.md file for more explanations on these files.

---

## [Editor Report · Decision Letter 1]

16 Aug 2024

PONE-D-24-09740R1Predefined and data-driven CT radiomics predict recurrence-free and overall survival in patients with pulmonary metastases treated with stereotactic body radiotherapyPLOS ONE

Dear Dr. Oikonomou,

Thank you for submitting your manuscript to PLOS ONE. After careful consideration, we feel that it has merit but does not fully meet PLOS ONE’s publication criteria as it currently stands. Therefore, we invite you to submit a second revised version of the manuscript that addresses the points raised during the review process.

**The authors should provide a clearer and more exhaustive response to Reviewer 1's first comment (i).**

**Please elaborate on which could be the actual reasons for C-values between 0.6 and 0.7 and which strategies could be enacted to improve them. In doing so, you should avoid vague terms such as 'notorious difficulty', 'detailed characteristics' (which ones? e.g. tumor biological features? enrollment criteria? patient history? treatment data? other?) and discuss, eventually with reference to the relevant literature and/or current practice.**

**I also suggest improving the English language and style of the manuscript, ideally with the aid of a native English-speaking medical writer or a professional editing service.**

We look forward to receiving your revised manuscript.

Kind regards,

Lorenzo Faggioni, M.D., Ph.D.

Academic Editor

PLOS ONE
---

## [Author Response · Author response to Decision Letter 1]

18 Sep 2024

We would like to thank the reviewers for their valuable comments. We have tried to the best of our ability to address these comments below. Please note that the Reviewer’s comments have been added in “italic” and any copied and pasted text from the revised manuscript in “Times New Roman”:

i) Regarding the C-index values obtained in this study, which range from 0.6 to 0.7, I think the authors should thoroughly discuss this point. It would be beneficial to explore potential strategies for enhancing these values. 

The authors should provide a clearer and more exhaustive response to Reviewer 1's first comment (i).

Please elaborate on which could be the actual reasons for C-values between 0.6 and 0.7 and which strategies could be enacted to improve them. In doing so, you should avoid vague terms such as 'notorious difficulty', 'detailed characteristics' (which ones? e.g. tumor biological features? enrollment criteria? patient history? treatment data? other?) and discuss, eventually with reference to the relevant literature and/or current practice.

Thank you for your insightful comment. We have now included a detailed presentation of our predictive performances (expressed as C-index values) with regards to the most recent literature on predictive models for metastases recurrence and patient survival. Moreover, we elaborated more on the potential most promising directions to improve the existing model performances. 

We have expanded the last paragraph of the Limitations section of the Discussion as follows (please see Revised Manuscript - without track changes: lines 588-568):

“Finally, the predictive performances of our best models (C-index: 0.66 and 0.67) for RFS and overall survival are in the range of recent similar studies. For the RFS predictions of patients with colorectal peritoneal metastases undergoing cytoreductive surgery and using clinical variables only, Dietz et al. [57] reported a C-index of 0.64. Sanli et al. [58] reported a predictive survival model with spinal bone metastases with a C-index of 0.669 (0.598–0.740) using composite clinical and radiomic score predictors. Recently, Liao et al. [59] presented a deep learning MRI radiomic model with a C-index of 0.75 to predict survival in patients with brain metastases treated with gamma knife surgery. Based on contrast-enhanced CT for brain metastases, Zhang et al [60] reported a C-index of 0.66 when using both clinical and radiomic predictors. Finally, using contrast-enhanced MRI images, Zhou et al [61] reported a C-index of 0.78 for clinical and volume-based models in patients with brain metastases who were undergoing stereotactic radiosurgery. These performances remain low for definite clinical applicability and should be considered as first steps toward future predictive survival models. Methodologic improvement for better recurrence and predictive survival models are suggested by the recent literature including large dataset prospective studies, homogeneous cancer origins [62], and deep-learning survival models following the predictor selection [59]. Moreover, a one-point radiomic and morphologic analysis ignores most of the dynamic of the cancer progression. Short-term radiomic and morphologic changes may be possible with accurate automatic segmentation, potentially adding predictive power without requiring long follow-up periods. This post-treatment serial follow-up approach may contribute to early recurrence prediction and may be combined with new genetic tests such as serial circulating tumour DNA (ctDNA), recently evaluated for predicting recurrence in colorectal cancer liver metastasis surveillance studies [63].”

ii) I also suggest improving the English language and style of the manuscript, ideally with the aid of a native English-speaking medical writer or a professional editing service.

Thank you for your valuable comment and recommendation. We have done our best to edit and improve the manuscript with the help of a professional service and we believe that the revised manuscript has been substantially improved.

---

## [Editor Report · Decision Letter 2]

23 Sep 2024

Predefined and data-driven CT radiomics predict recurrence-free and overall survival in patients with pulmonary metastases treated with stereotactic body radiotherapy

PONE-D-24-09740R2

Dear Dr. Oikonomou,

We’re pleased to inform you that your manuscript has been judged scientifically suitable for publication and will be formally accepted for publication once it meets all outstanding technical requirements.

Kind regards,

Lorenzo Faggioni, M.D., Ph.D.

Academic Editor

PLOS ONE

---

## [Editor Report · Acceptance letter]

30 Sep 2024

PONE-D-24-09740R2 

PLOS ONE

Dear Dr. Oikonomou, 

I'm pleased to inform you that your manuscript has been deemed suitable for publication in PLOS ONE. Congratulations! Your manuscript is now being handed over to our production team.

Kind regards, 

on behalf of

Dr. Lorenzo Faggioni 

Academic Editor

PLOS ONE